# Determining *Pneumocystis jirovecii* Colonisation from Infection Using PCR-Based Diagnostics in HIV-Negative Individuals

**DOI:** 10.3390/diagnostics14010114

**Published:** 2024-01-04

**Authors:** Anna Louise Watson, John Woodford, Sumudu Britton, Rita Gupta, David Whiley, Kate McCarthy

**Affiliations:** 1Infectious Diseases, Royal Brisbane & Women’s Hospital, Metro North Health, Herston, QLD 4006, Australia; 2Herston Infectious Diseases Institute, Herston, QLD 4006, Australia; 3Infectious Diseases, Ipswich Hospital, Ipswich, QLD 4305, Australia; 4Pathology Queensland, Herston, QLD 4006, Australia; 5The University of Queensland, Herston, QLD 4006, Australia

**Keywords:** quantitative PCR, colonisation, diagnostics, immunocompromised hosts, *Pneumocystis jirovecii*, pneumonia

## Abstract

Background: *Pneumocystis jirovecii* pneumonia is increasingly diagnosed with highly sensitive PCR diagnostics in immunocompromised, HIV-negative individuals. We assessed the performance of our in-house quantitative PCR with the aim to optimise interpretation. Methods: Retrospective audit of all positive *P. jirovecii* qPCRs on induced sputum or BAL fluid at a single centre from 2012 to 2023. Medical and laboratory records were analysed and people with HIV were excluded. Cases were categorised as colonisation, high-probability PCP or uncertain PCP infection against a clinical gold standard incorporating clinico-radiological data. Quantitative PCR assay targeting the 5s gene was utilised throughout the time period. Results: Of the 82 positive qPCRs, 28 were categorised as high-probability PCP infection, 30 as uncertain PCP and 24 as colonisation. There was a significant difference in qPCR values stratified by clinical category but not respiratory sample type. Current assay performance with a cutoff of 2.5 × 10^5^ copies/mL had a sensitivity of 50% (95% CI, 30.65–69.35%) and specificity of 83.33% (95% CI, 62.62–95.26%). Youden Index calculated at 6.5 × 10^4^ copies/mL had a sensitivity of 75% (56.64–87.32%, 95% CI) and specificity of 66.67% (46.71–82.03%, 95% CI). High and low cutoffs were explored. Significant variables associated with infection were age > 70 years old, the presence of fever, hypoxia or ground glass changes. Conclusions: A single qPCR cutoff cannot reliably determine *P. jirovecii* infection from colonisation. Low and high cutoffs are useful, however, a large “possible infection” cohort will remain where interpretation of clinic-radiological factors remains essential. Standardisation of assays with prospective validation in specific immunocompromised groups will allow greater generalisability and allow large-scale prospective assay validation to be performed.

## 1. Introduction

*Pneumocystis jirovecii* is a fungal organism that can transiently colonise the human respiratory tract and progress to life-threatening pneumonia in susceptible hosts. While historically seen in people with human immunodeficiency virus (HIV), there is an increasing incidence in people with malignancy, organ transplantation and autoimmune disease [1]. *P. jirovecii* pneumonia (PCP) diagnosis requires various clinical, radiological and microbiological criteria to be fulfilled. While an immunocompromising state is required for the diagnosis of invasive fungal disease, this host factor, while expected, is not required to meet the diagnostic criteria of PCP. If a predisposing factor is not evident, investigations for an underlying immunodeficiency are recommended [2]. The 2020 Update EORTC/MSGERC Definitions of Invasive Fungal Diseases clarified definitions of proven and probable PCP in individuals without HIV [2]. Identification of *P. jirovecii* by microscopy is required to fulfil the microbiological criteria. Quantitative PCR is included in the probable PCP criteria and is yet to be accepted as part of the proven PCP definition given the lack of standardised methodology and interpretation of values for distinguishing infection from colonisation [2]. Microbiological testing alone cannot decipher between these categories, acknowledging that between 15.9% to 58.8% of immunocompromised hosts (ICHs) will be colonised with *Pneumocystis* [3,4]. Colonisation can lead to propagation in immunosuppressed cohorts and outbreaks. Given the acute and rapidly progressive nature of PCP in ICHs, with higher rates of intensive care admissions and mortality compared to their HIV-positive counterparts, nuanced and timely diagnostics are required.

Microbiological diagnosis of *P. jirovecii* has advanced over the past several decades. Identification of the organism from respiratory samples was traditionally achieved with microscopy enhanced with tinctorial (dye-based) staining or fluorescent antibody staining which requires skilled interpretation. The sensitivity of microscopy with staining on induced sputum is 50–90% and over 90% on BAL fluid in people with HIV acknowledging a lower yield in HIV-negative individuals due to fewer cysts and more inflammatory cells [5,6]. PCR-based diagnostics are exceedingly sensitive with a documented 100% sensitivity and negative predictive value when validated against cytology and surgical pathology [7].

Acknowledging that the disease process and fungal burden inciting PCP are different in HIV-positive and negative individuals, PCR-based diagnostics require validation in each group. While highly sensitive testing is advantageous for ICHs who are known to have a lower fungal burden causing PCP [8], specificity is compromised and the inability to determine colonisation from infection becomes problematic. Diagnostic performance on different respiratory specimens also needs to be taken into account given testing can be performed on a range of samples including induced sputa, bronchoalveolar lavage fluid and oropharyngeal washings.

We aim to characterise the population with a positive *P. jirovecii* qPCR and stratify infection and colonisation using a clinical gold standard. We sought to evaluate the performance of our current in-house qPCR *P. jirovecii* assay targeting the 5s gene on lower respiratory specimens in HIV-negative individuals with the aim of optimising the utility of PCR-based diagnostics.

## 2. Materials and Methods

We performed a single-centre retrospective audit of all patients with positive *P. jirovecii* PCRs at the Royal Brisbane and Women’s Hospital, a 986-bed hospital in Australia. The current assay in use was implemented in February 2012 so data were collected from that time to February 2023. Individuals with *P. jirovecii* detected on induced sputum or bronchoalveolar lavage fluid (BALF) were identified from laboratory records and clinical records retrieved.

### 2.1. Data Collection

We collected demographic factors (age, sex), medical factors (malignancy, transplantation), chronic lung (chronic obstructive airways disease or pulmonary fibrosis) or liver disease, chronic kidney disease requiring dialysis, diabetes requiring medication, diabetes and concurrent steroid use equivalent of ≥15 mg prednisolone per day for 2 or more weeks, autoimmune or connective tissue disease, prescribed immunosuppression and PCP prophylaxis), clinical factors (the presence of fever, dyspnoea, cough or hypoxia < 95% on room air) and radiological factors (X-ray or computed tomography imaging with radiologist reporting ground glass changes, nodules or consolidation) to characterise the population. Details of chemoprophylaxis adherence and duration were not assessed. Concurrent infections (bacterial, viral, fungal) and alternate diagnoses contributing to the individual’s presentation if recorded in patient notes or discharge summaries were identified to allow accurate pneumocystis categorisation. The specimen type (BALF, induced sputa) and quantitative PCR value were recorded. If multiple samples for an individual existed during the same illness, the BALF sample was selected and if multiple BALF samples existed, the high qPCR value was recorded. Clinician-initiated treatment and mortality at day 30 from positive PCR results were recorded. Receiving immunosuppression was defined as being prescribed a steroid equivalent of ≥15 mg prednisolone per day for 2 or more weeks, any form of chemotherapy, immunotherapy, conventional synthetic disease-modifying anti-rheumatic drugs (csDMARDs) or biologic therapies within the past month. An individual could be included more than once if the respiratory specimen was collected during a new clinical illness. People with HIV or those aged less than 18 years old were excluded.

Individuals were classified as high-probability PCP if all of the following criteria were met:

(i) Compatible presentation (presence of 1 or more: fever, cough, dyspnoea, hypoxia); (ii) compatible radiological findings (presence of 1 or more: ground glass changes, nodules, consolidation); (iii) PCP treatment commenced; and (iv) no contributing concomitant diagnoses identified (infective or non-infective). Criteria (iii) were fulfilled if treatment was intended but the patient died before receiving therapy.

Individuals were classified as uncertain PCP if criteria (i) and (iii) were met but not (ii) or (iv). If individuals were not treated then they were classified as colonisation. This definition takes into account clinic–radiological and microbiological factors described in the EORTC guidelines^2^, although it differs in that clinician-initiated treatment influenced categorisation and that cases with contributing factors were downgraded from “high-probability PCP” to “uncertain PCP” infection to ensure confidence in the clinical gold standard of infection. PCR diagnostics are the only routinely used test at our institution for identifying *P. jirovecii* so other microbiological data were not collected.

### 2.2. PCP PCR Assay

Pathology Queensland developed an in-house *P. jirovecii PCR* assay in 2012 using primers (forward 5′-AGTTACGGCGATACCTCAGAGAATATAC-3′ and reverse 5′-GCTACAGCACGTCGTATTCCCATA-3′) and a probe (5′-FAM-TCACCCACTATAGTACTGACGACGCCCTT-BHQ1-3′) targeting the 5s rRNA gene [9]. The samples were extracted on Roche MagNApure 96 instrument using 200 µL of sample and eluted in 100 µL of elution buffer. The master mix is made using Qiagen Quantitech probe master mix with 10 pmoles/µL of *P. jirovecii* primers and 20 pmoles/µL of the probe in a 20 µL reaction along with 5 µL of clinical sample extract. All requests for *P. jirovecii* PCR are screened by qualitative PCR with a CT value cutoff of 40 (this equates to a lower limit of detection (LLD) qPCR of 8 × 10^3^ copies/mL (CT = 39) to 8 × 10^2^ copies/mL (CT = 42)). The *P. jirovecii* positive samples were then run with *P. jirovecii* standards (made in house using positive clinical samples) with known copies/mL (8.00 × 10^4^ to 8.00 × 10^7^) and the runs were performed on Qiagen Rotorgene Q real-time PCR instrument and analysed with Rotorgene Q software. The “current” qPCR cutoff indicative of infection generated in an HIV cohort was established based on in-house testing on unpublished data over 15 years ago. This was established by assessing a small number of respiratory samples from HIV-positive patients with positive and negative microscopy for *P. jirovecii* and establishing a discriminating qPCR cutoff value.

## 3. Analysis

Study population characteristics stratified by clinical diagnosis were presented descriptively as proportions for categorical variables or median, interquartile range (IQR) for continuous variables. Population characteristics were compared between the high-probability PCP and colonised groups using the Fisher exact test (categorical variables) and Mann–Whitney U test (continuous variables). High-probability PCP, uncertain PCP and colonised clinical diagnosis groups were compared using the Chi-squared test (categorical variables) and Kruskal–Wallis test (continuous variables). No corrections have been made for multiple comparisons.

Quantitative PCR results are presented as median, IQR and visualized using a log10 scale and stratified by sample type and clinical diagnosis. BALF versus sputum results were compared using the Mann–Whitney U test.

To understand the real-world utility of the current qPCR assay cutoff, we evaluated assay sensitivity and specificity relative to a clinical diagnosis gold standard in the colonised and high-probability PCP PCP groups. Concordance between assay positivity using the existing cutoffs and clinical diagnosis was assessed using Cohen’s kappa.

A ROC curve was generated using the high-probability PCP, and colonised groups and cutoff optimisation were evaluated by two methods. Firstly, a cutoff balancing sensitivity and specificity was generated using the Youden index to provide the maximised balance between sensitivity and specificity. Concordance between assay positivity using the Youden cutoffs and clinical diagnosis was assessed using Cohen’s kappa. Secondly, a two-cutoff system was evaluated using a low cutoff maximising sensitivity and a low cutoff maximising specificity. To understand how optimisation might affect clinical practice, the uncertain PCP group was re-classified using the old and proposed cutoffs.

To understand factors influencing clinical diagnosis an exploratory multivariate analysis was performed. The effects of selected covariates were modelled by logistic regression using the clinical diagnosis outcomes ‘high-probability PCP’ or ‘colonised’. Age, (<50, 50–70, >70 years), sex, prophylaxis, sample type and qPCR result were included a priori. Fever and ground glass changes were included based on the univariate analysis using a threshold *p* < 0.05.

Analysis was performed using GraphPad Prism v9. This study was approved with a waiver of patient consent by the Metro North Health Human Research Ethics Committee (Project ID: 81031).

## 4. Results

A total of 82 positive *P. jirovecii* PCRs were detected between February 2012 and February 2023; 28 cases were categorised as high-probability PCP infection, 30 as uncertain PCP infection and 24 as colonisation. Patient characteristics stratified by clinical diagnosis are presented in Table 1.

### 4.1. Study Population and Clinical Characteristics

The majority of patients were male (67% (55/82)) with a median age of 62 years old (IQR 54–71.75). The most frequent co-morbidities included active malignancy (72% (59/82)) and chronic lung disease (39% (32/82)) and 84% of patients (69/82) were prescribed some form of immunosuppression. (Figure 1). PCP prophylaxis was prescribed in 13% (11/82) of patients. Overall, dyspnoea was the most frequent clinical presentation (87% (71/82)) followed by cough (68% (56/82)), hypoxia (65% (53/82)) and fever (45% (37/82)). When stratified by clinical diagnosis, hypoxia and fever were least common in the colonisation group compared to the uncertain PCP and high-probability PCP groups (42% (10/24) vs. 80% (24/30) vs. 68% (19/28), *p* = 0.0125 and 21% (5/24) vs. 47% (14/40) vs. 64% (18/28), *p* = 0.0071, respectively). Radiographic abnormalities were common with ground glass changes (51% (42/82) and consolidation (48% (39/82)) the most frequently reported. Compared to the colonised group, ground glass changes were more common in the uncertain PCP and high-probability PCP groups (21% (5/24) vs. 60% (18/30) vs. 68% (19/28), *p* = 0.0016. The 30-day mortality in the colonised, uncertain PCP and high-probability PCP groups was 13% (3/24), 27% (8/30) and 18% (5/28), respectively (*p* = 0.4111). Notably, 2 patients in the uncertain PCP group died before PCP treatment could be commenced. All patients in the study had at least 1 symptom compatible with PCP and 1 radiological abnormality, regardless of group classification.

### 4.2. P. jirovecii Sampling and qPCR Results

Of the 82 positive PCRs, 53 were from induced sputa and 29 were from BALF samples. BALF sampling was performed for 25% (6/24), 43% (13/30) and 36% (10/28) of the colonised, uncertain PCP and high-probability PCP patients, respectively (*p* = 0.3748). The median (IQR) *P. jirovecii* qPCR value was highest in the high-probability PCP group (2.8 × 10^5^ copies/mL (IQR 6.375 × 10^4^–2.525 × 10^6^ copies/mL)), followed by the uncertain PCP group (1.015 × 10^5^ copies/mL (IQR 1.65 × 10^4^–8.85 × 10^5^ copies/mL)) and lowest in the colonised group (4.05 × 10^4^ copies/mL (IQR 1.18 × 10^4^–1.475 × 10^5^ copies/mL)), *p* = 0.0093) (Figure 2). While sputum and BALF sample qPCR results were similar for the colonisation and high-probability PCP groups, in the uncertain PCP group, there was a difference in qPCR results based on sample ((BALF 1.5 × 10^4^ copies/mL (IQR 8 × 10^3^–2.8 × 10^5^ copies/mL)) and sputum ((1.4 × 10^5^ copies/mL (IQR 4.9 × 10^4^–9.9 × 10^5^ copies/mL), *p* = 0.0481), (Figure 3).

### 4.3. Assay Performance and Cutoff Optimisation

The performance of the current qPCR-derived cutoff of 2.5 × 10^5^ copies/mL was assessed using the colonised and high-probability PCP infection cohorts. ROC curve generated using the colonised and high-probability PCP infection cohorts gave an area under the curve (AUC) of 0.7545 (95% CI 0.6248–0.8841) (Figure 4A). Applying the current qPCR-derived cutoff of 2.5 × 10^5^ copies/mL generated in an HIV cohort, assay sensitivity was 50% (95% CI 30.65–69.35) and specificity 83.33% (95% CI 62.62–95.26). This cutoff classes 11/30 in the uncertain PCP infection group above the threshold. The Youden Index generated an optimal cutoff value of 6.5 × 10^4^ copies/mL, resulting in an assay sensitivity of 75% (95% CI 56.64–87.32) and specificity of 66.67% (95% CI 46.71–82.03) (Figure 4B). This cutoff classes 16/30 in the uncertain PCP infection group above the threshold.

The concordance between clinical (high-probability PCP or colonised) and laboratory (positive or negative) was fair for the existing cutoffs (Kappa = 0.324) and moderate (Kappa = 0.418) for the optimised cutoff.

A two-threshold system was explored. A high assay cutoff >1.55 × 10^6^ copies/mL has 100% specificity and a sensitivity of 32% (17.95–50.66%). A low assay cutoff off >8.1 × 10^3^ copies/mL has a sensitivity of 96.43% (95% CI 82.29–99.82) and specificity of 20.83% (95% CI 9.245–40.47), noting that the LLD of the assay is 8 × 10^2^ copies/mL to 8 × 10^3^ copies/mL. The bigger the difference between the two selected cutoffs, the bigger the grey zone.

### 4.4. Exploratory Analysis

The effects of selected covariates on clinical diagnosis (colonised or high-probability PCP infection) were modelled by multiple logistic regression. Using the co-variates age, sex, prophylaxis, fever, ground glass changes, sample type and qPCR, the model AUC was 0.9315 (95% CI 0.8578–1.000) (Figure 4B). The qPCR result was not independently associated with the clinical diagnosis (OR 1.97, 95% CI 0.7528–6.579). Male sex (OR 0.14, 95% CI 0.01201–0.9536), age > 70 years (OR 69.04, 95% CI 4.179–3054), ground glass changes on imaging (OR 22.5, 95% CI 2.55–554.4) and BALF sample collection (OR 20.29, 95% CI 2.166–548.5) were independently associated with clinical diagnosis.

## 5. Discussion

In this single-centre study, we evaluated the clinical, radiographic and laboratory features of non-HIV patients with positive *P. jirovecii* PCR testing using our local standardised assay. Using a clinical gold standard, the current quantitative PCR test cutoff sensitivity of 50% is suboptimal in an infection associated with high mortality [10]. The low sensitivity identified supports the observation that HIV-negative ICHs may have a lower organism burden than their HIV-positive counterparts [11]. In this study, common comorbidities included malignancy, chronic lung disease and autoimmune/connective tissue disease which is in keeping with the rising prevalence of PCP in various HIV-negative immunocompromised groups [1]. The relatively low proportion of cases in haematology patients may be due to the frequent protocolized use of chemoprophylaxis at our institution, which is expected to be highly efficacious in preventing PCP and may reflect a further shift in PCP epidemiology.

Given the shifting epidemiology of PCP, it is critical to optimise testing for use in an increasingly non-HIV target population. Our current assay performance is modest, with an AUC of 0.7545. While a negative test is useful for excluding PCP, a positive test alone does not discriminate between infection and colonisation without consideration of the clinical picture. In our study cohort, attempts to optimise a single cutoff resulted in improved sensitivity, but a trade-off in specificity increased the risk of overtreatment. Utilising ‘high’ and ‘low’ threshold cutoffs created a large grey zone of “uncertain PCP infection”, leaving the decision to initiate treatment at the discretion of the treating clinician. This is similar to the findings of several other studies in which the “uncertain PCP” cohort qPCR value fell between the values found in the high-probability PCP and colonised groups [12]. As a result, we identified variables that may alter the pre-test probability and assist with clinical qPCR interpretation. In this cohort, the most useful variables associated with infection were an age > 70 years old, the presence of fever, hypoxia or ground glass changes. ICHs identified with *P. jirovecii* colonisation can be considered for chemoprophylaxis, which may prevent the development of PCP.

Our study has several limitations. As a single-centre study at a facility that sees relatively few solid organ transplant recipients, this at-risk group is underrepresented. Larger and truly generalisable studies will continue to be challenging until a commonly accepted clinical definition and reference assay are widely available, resulting in multiple smaller local studies such as ours [12,13,14]. Furthermore, the use of a clinical gold standard which was determined by the treating clinician is inherently subjective. While the decision to initiate treatment may have been influenced by the assay results, qPCR was not independently associated with clinical diagnosis in multivariate analysis, suggesting other factors influence clinician practice. Through the meticulous collection of host, clinical, radiological and microbiological data including establishing alternate contributing factors, we have attempted to follow the most accurate retrospective clinical definition of PCP. A variety of definitions for establishing PCP infection have been utilised in the literature highlighting the wider diagnostic issue, which has been somewhat addressed by the revised EORTC/MSGERC PCP definitions [2]. We chose to use different terms, high-probability PCP and uncertain PCP, to avoid confusion with the EORTC definitions as our groups differ from the guidelines in terms of categorisation.

The net state of immunosuppression of the host plays an important role in the interpretation of a positive PCP PCR. Ideally, we would have defined and analysed the interplay between the degree of immunodeficiency and the significance of the qPCR in relation to infection and colonisation. What is recognised as a significant qPCR value may be different depending on the degree of cell-mediated and humoral immune deficiency between hosts. The study size is too small to perform subgroup analysis on those with significant immunodeficiency alone.

Finally, sampling bias may exist and participants did not have matched BAL and induced sputa specimens. Ideally, all samples would have had microscopy using conventional or immunofluorescence staining performed to assess the correlation between the detection of *P. jirovecii* and clinical diagnosis. The timing of sample collection during illness duration and its relationship to the commencement of treatment will also influence the qPCR value. BAL and induced sputa qPCRs were found to be comparable in our colonised and high-probability PCP groups; however, this is not consistent across the literature and guidelines [11,15] and likely reflects a selection bias for procedure chosen and non-standardised BALF collection across centres. This was most pronounced in our uncertain PCP group, where 13 individuals with a median BAL qPCR lower than the median qPCR in the colonised cohort were treated for PCP, suggesting that if the clinician had a high enough index of suspicion to test, they took any result as positive.

Finally, while we have generated new qPCR cutoffs, these have not been externally validated or generalisable. Comparison between assays has been performed with significant variability depending on the target gene and qPCR method/template used [16]. Given the modest performance of improved assay cutoffs, even if consistent across populations, the current test performance may not substantially improve diagnostic confidence.

While the qPCR remains a valuable tool, consideration of clinico-radiological factors remains essential in the diagnosis of PCP. Clinical and laboratory harmonization is needed before a prospective multicentre study can reasonably overcome the current issues of generalisability and subgroup heterogeneity. As a result, in the absence of a clear single test cutoff, multimodal risk calculators and adjunctive tests such as Beta-D-glucan [12,17] may warrant further investigation for inclusion in future diagnostic algorithms to ascertain the most accurate diagnosis of PCP.

## Figures and Tables

**Figure 1 diagnostics-14-00114-f001:**
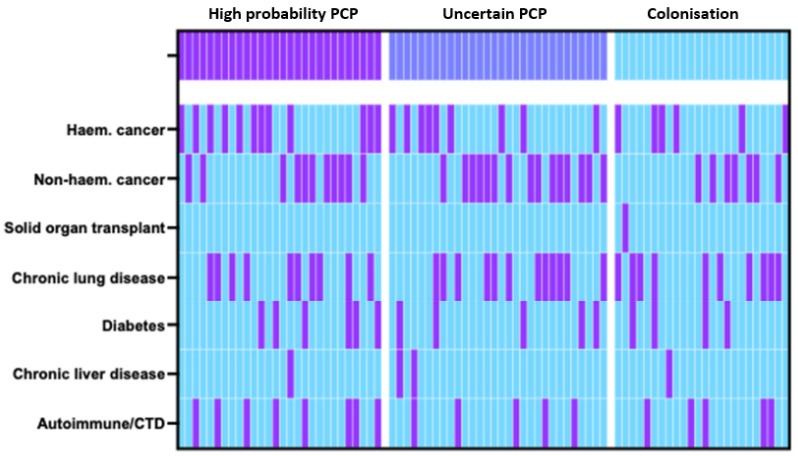
Heatmap of study population comorbidities stratified by clinical diagnosis. Each vertical column represents 1 individual, with purple indicating present and blue indicating absent.

**Figure 2 diagnostics-14-00114-f002:**
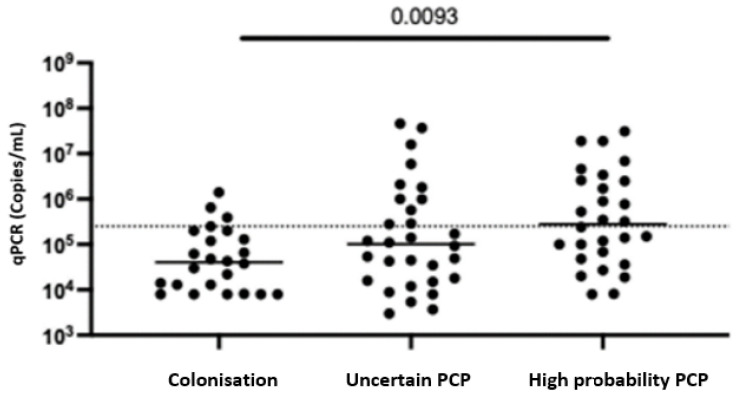
qPCRs on any sample stratified by diagnosis. Bars at median. Kruskall–Wallis Test. Dotted line is existing cutoff.

**Figure 3 diagnostics-14-00114-f003:**
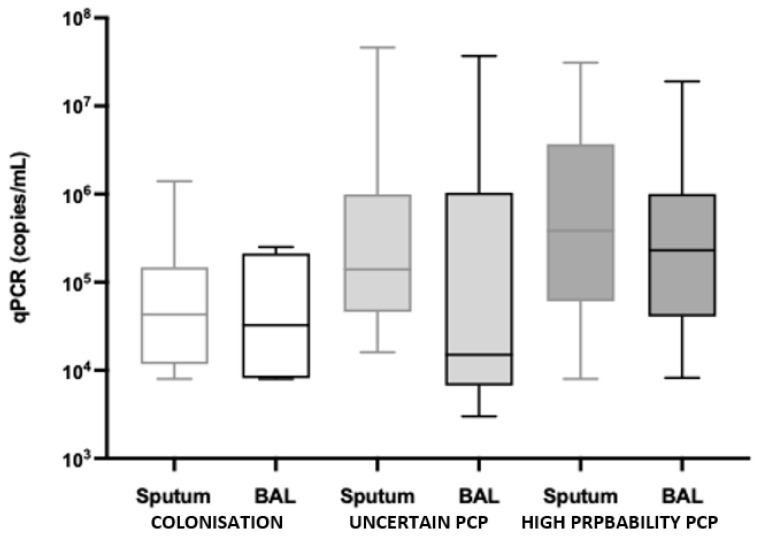
qPCR value stratified by diagnosis and sample type.

**Figure 4 diagnostics-14-00114-f004:**
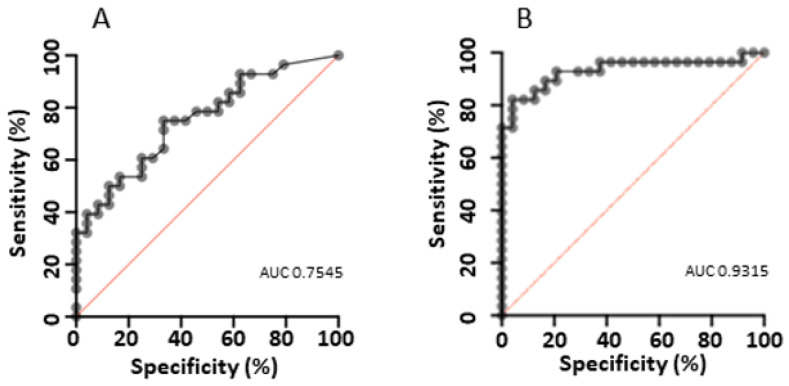
(**A**,**B**) ROC curves for the differentiation of Pneumocystis high-probability PCP infection and colonisation using (**A**) qPCR values alone (**B**) multivariate model of clinical, radiographic and laboratory factors (age, sex, prophylaxis, fever, ground glass changes, sample type and qPCR).

**Table 1 diagnostics-14-00114-t001:** Demographics stratified by clinical diagnosis.

Characteristics	Colonisation (*n* = 24)	Uncertain PCP (*n* = 30)	High-Probability PCP (*n* = 28)	3-Way Comparison	2-Way Comparison (High-Probability PCP vs. Colonised)
Sex (male)	70.8% (17/24)	66.6% (20/30)	57.1% (16/28)	0.5641	0.3911
Age (median, IQR)	57 (42–66)	68 (62–73)	63 (53–74)	0.0192	0.1137
Sample type (BAL)	25% (6/24)	43.3% (13/30)	35.7% (10/28)	0.3748	0.5487
qPCR (median, IQR)	40,500 (11,800–147,500)	101,500 (16,500–885,000)	280,000 (63,750–2,525,000)	0.0093	0.0013
BAL qPCR	32,500 (11,650–160,750)	15,000 (8000–280,000)	230,000 (66,000–665,000)	0.1913	0.0978
Sputum qPCR	43,000 (13,000–127,500)	140,000 (49,000–990,000)	385,000 (76,750–3,200,000)	0.0113	0.0056
Co-morbidities Haem. cancer	25% (6/24)	30% (9/30)	42.9% (12/28)	0.3589	0.2452
Non-haem. cancer	29.2% (7/24)	50% (15/30)	39.3% (11/28)	0.2943	0.4615
Solid organ transplant	4.2% (1/24)	0% (0/30)	0% (0/28)	0.2978	0.5622
Chronic lung disease	41.7% (10/24)	40% (12/30)	35.7% (10/28)	0.8997	0.7772
Diabetes	16.7% (4/24)	16.7% (5/30)	21.4% (6/28)	-	>0.9999
Diabetes and concurrent steroid use	25% (1/4)	20% (1/5)	66.7% (4/6)		
Chronic liver disease	4.2% (1/24)	6.7% (2/30)	3.6% (1/28)	0.8453	>0.9999
Autoimmune/CTD	20.8% (5/24)	16.7% (5/30)	28.6% (8/28)	0.5426	0.749
Immunosuppression medications	58.3% (14/24)	80% (24/30)	67.9% (19/28)	0.2222	0.5685
Steroids ≥ 2 wks > 15 mg/day	25% (6/24)	36.7% (11/30)	28.6% (8/28)	0.6281	>0.9999
Prophylaxis	8.3% (2/24)	13.3% (4/30)	17.9% (5/28)	0.6037	0.43
Clinical presentation Fever	20.8% (5/24)	46.7% (14/30)	64.3% (18/28)	0.0071	0.0022
Dyspnoea	75% (18/24)	96.7% (29/30)	85.7% (24/28)	0.0666	0.4829
Cough	66.7% (16/24)	73.3% (22/30)	64.3% (18/28)	0.7449	>0.9999
Hypoxia (<95%)	41.7% (10/24)	80% (24/30)	67.9% (19/28)	0.0125	0.0926
Bacterial	12.5% (3/24)	40% (12/30)	0% (0/28)	0.0003	0.0916
Viral	20.8% (5/24)	30% (9/30)	0% (0/28)	0.0085	0.0164
Other infective	16.7% (4/24)	10% (3/30)	0% (0/28)	0.0941	0.0393
Non-infective	75% (18/24)	53.3% (16/30)	0% (0/28)	<0.0001	<0.0001
Radiological findings					
Ground glass	20.8% (5/24)	60% (18/30)	67.9% (19/28)	0.0016	0.0009
Reticular opacities	0% (0/24)	3.3% (1/30)	3.6% (1/28)	0.6532	>0.9999
Nodules	25% (6/24)	20% (6/30)	21.4% (6/28)	0.9042	>0.9999
Consolidation	45.8% (11/24)	40% (12/30)	57.1% (16/28)	0.4175	0.5783
Deceased D+30	12.5% (3/24)	26.7% (8/30)	17.9% (5/28)	0.4111	0.7109
PCP treatment	0% (0/24)	93.3% (28/30)	100% (28/28)	<0.0001	<0.0001

## Data Availability

The data presented in this study are available in the Appendix A.

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
