# Peer review of "Determining Pneumocystis jirovecii Colonisation from Infection Using PCR-Based Diagnostics in HIV-Negative Individuals"

_diagnostics, 2024, doi:10.3390/diagnostics14010114_

Round 1
Reviewer 1 Report
Comments and Suggestions for Authors
Watson and colleagues aimed to set a cut-off for PJP load, in which differentiation between colonization or (probable or proven) infection could be established. This research is relevant and conducts to the knowledge on how PJP load is related to the infection status. So far, a true cut-off could not be established. Although this might not be novel research, Watson and colleagues tried to fulfill this knowledge gap.
I believe that this research is well-performed and the paper is clear and well written. The authors included all PJP positive patients during a very long time period. Although the study period was 11 years, only 82 patients could be included. A bigger study consortium would give this research more power, which has also been discussed in the paper.
Authors have performed proper statistics and results are presented in a clear way. The figures and tables contribute to the clearness of the paper.
I have no comments on this well written and clear paper.
Author Response
Thank you for your review and positive response to the paper.
Reviewer 2 Report
Comments and Suggestions for Authors
This manuscript by Watson, et al. is a retrospective single-center study of the diagnostic utility of quantitative P. jirovecii PCR in 82 patients without HIV with a positive qualitative PCR result over an 11-year period. The authors assign a "gold standard" label of proven infection, probable infection, or colonisation based on their own definitions which draw on EORTC/MSGERC consensus definitions but deviate in important ways, most notably with the absence of microscopy in the current study, which is the singular criteria for proven infection in the EORTC/MSGERC definitions, with omission of host factors from their gold standard criteria, and with inclusion of the clinical decision whether or not to treat as a requirement for this study's definition of "proven infection." The diagnostic utility of quantitative PCR in distinguishing the "proven infection" from "colonization" group is described by ROC curves and various cutoff values are described. The analysis is complemented by a multivariate logistic regression analysis including quantitative PCR result and clinical factors which did not show the quantitative PCR result to be associated with the gold standard. Overall, I feel this study is a useful contribution to the limited number of studies addressing this topic and I expect it to be suitable for publication in Diagnostics with certain revisions. My biggest concern is with the gold standard definitions. At a minimum, use of identical terminology but different criteria compared to the EORTC/MSGERC consensus definition is confusing. Use of treatment decision in the criteria and excluding this study's "probable" group also seems problematic. My specific suggestions are below:
Abstract
Line 30: "possible" is misspelled
Introduction
The authors should revise their statement about "clinical, radiological, and microbiological criteria" and "immunocompromising state is not required" in lines 44-45. EORTC/MSGERC revision of the consensus definitions (https://doi.org/10.1093/cid/ciz1008 Table 2 footnote) states "Probable invasive fungal diseases (IFD) requires the presence of at least 1 host factor, a clinical feature and mycologic evidence and is proposed for immunocompromised patients only, whereas proven IFD can apply to any patient, regardless of whether the patient is immunocompromised." So proven IFD requires only microscopic detection and does not require an immunocompromised state, whereas probable IFD (which is what relies on characteristics from multiple domains to be present) does require at least one host factor that implies an immunocompromised state.
Methods
Line 90 change "was" to "were".
Please provide further clarification of which immunotherapies and biologics were considered to be immunosuppression would be helpful both in interpreting the study. I suggest including the specific agents in the patient-level data in the supplement. Are the authors sure that all of the immunotherapies and biologics are associated with T cell dysfunction?
My chief concern with the paper is in the criteria used to define the gold standard of proven/probable/colonization. Use of the terms "Proven PCP" and "Probable PCP" but with definitions that differ significantly from the EORTC/MSGERC definitions is confusing. I am concerned that there are no host criteria included in the definition, so it is not clear to me whether all the patients in this study meet "Host factor" criteria as outlined in EORTC/MSGERC definitions. Also, although I understand the author's rationale, the use of treatment decision and the complete exclusion of the "probable" group from analysis is also problematic because there would be a tendency to treat in certain scenarios even if true clinical suspicion for PCP is low and an alternate diagnosis does not exclude concurrent PCP, especially in an immunocompromised host.
I suggest revising the terms used to describe the author's current categories so that they are distinct from the terms used in the EORTC/MSGERC consensus definitions. I also suggest adding an analyiss that follows EORTC/MSGERC consensus definitions strictly. The authors do not have any "proven" cases by EORTC/MSGERC definitions because microscopy data are not included. So the included patients would be divided into "probable infection" or "colonized." This would ensure that all the included patients are immunocompromised, eliminate the problem of excluding more than a third of the data set from the subsequent analysis and in my opinion this version of the analysis would be more intrepretable to clinicians. I suspect the authors would still reach the same overall conclusions.
2.5e5 is described as the "current" cutoff generated in an HIV cohort, but no reference or explanation is provided as to where this came from. Since there is no consensus cutoff or assay standardization for Pneumocystis PCR, the authors should explain why this was treated as a meaningful cutoff to include (or exclude this cutoff from the analysis altogether).
Results
Overall, more detailed presentation of what makes the patients in this study immunocompromised would be helpful. The presentation of co-morbidities and aspects of immunocompromising conditions could be improved in table 1. I suggest separating out reasons for immunocompromise from other co-morbidities. Include rows in the table for immunotherapies, biologics, DMARDs, etc. underneath "immunosuppression medications" so the reader can better understand your population.
Similarly, what "chronic lung diseases" are included? Are all the patients with diabetes on steroids (and presumably the steroids contributed to their diabetes), or do you have patients in the study for whom diabetes is the only medical feature? (in which case they would not meet host factor criteria for the EORTC/MSGERC definitions).
Figure 1 would be more useful if it was revised to include the factors that predispose a host to PCP. A row for hemodialysis seems unnecessary, especially as most would not consider hemodialysis to be a condition that predisposes a host to PCP.
Figure 4A caption: I am confused by "qPCR with the current cutoff 2.5x10^5 copies/mL". The ROC curve is generated by calculating sensitivity and specificity throughout the range of potential cutoffs. Did the authors intend to highlight the data point on this ROC curve corresponding to a cutoff of 2.5x10^5 copies/mL?
Figure 4. How is the p value calculated? Please describe in the figure caption and methods.
Discussion
Line 306 - did the authors mean adjunctive rather than adjuvant?
Lack of cytology data should be discussed as a limitation of the study.
Author Response
Thank you Reviewer 2 for your in depth review of our manuscript. We have responded as best as able below.
Abstract
- Line 30: "possible" is misspelled – spelling corrected
Introduction
- Introduction re-written. Discussed that while a host factor is required for the diagnosis of Invasive Fungal Disease, the Pneumocystis jirovecii Disease: Basis for the Revised EORTC/MSGERC Invasive Fungal Disease Definitions in Individuals Without Human Immunodeficiency Virus consensus does not require a host factor for the diagnosis of PCP. I have acknowledged that we were unable to adhere to the EORTC/MSGERC Consensus Definitions given pneumocystis microscopy is rarely performed at our centre and there is a trend towards PCR replacing microscopy due to the improved sensitivity, especially in HIV negative patients with lower fungal burdens.
Methods
- We acknowledge that an indepth analysis of the immunosuppression and biologics prescribed is relevant and of interest. Unfortunately data was insufficient to accurately ascertain dosing, duration and adherence to prescribed immunosuppression as well as the effect on T cell function. Given the inclusion criteria was based on a positive qPCR and an immunocompromised host state was not required to be included in the study, we have recorded the prescription of immunosuppression of any form more to capture patient demographics rather than to assess risk associated with certain prescribed medications. Identified immunosuppressive medications were documented in the supplementary material however Diagnostics has asked us to remove the supplementary material file.
- To avoid confusion between the PCP EORTC/MSGERC Consensus Definitions and our study definitions, I have changed our categories from “proven” to “definite” and “probable” to “possible”.
- We acknowledge that categorising patients as either probable infection or colonisation would simplify interpretation and increase numbers to analyse however we have chosen to keep the 3 groups. We wanted to exclude patients that were treated for PCP but were unlikely to have true infection from our analysis to try to establish a more accurate qPCR cutoff for infection.
- The "current" cutoff generated in an HIV cohort was established based on in-house testing on unpublished data over 15 years ago. This was established assessing a small number respiratory samples from HIV positive patients with positive and negative microscopy for jirovecii and establishing a discriminating qPCR cutoff value. Despite the microbiology lab reporting a recommended cut-off indicative of infection, all positive qPCRs are reported and available to clinicians to interpret. An aim of our study was to provide a more robust assessment of assay performance, especially in the HIV-negative cohort in which it is primarily used.
- I have removed haemodialysis from Table 1 and Figure 1.
- Line 90: was changed to were
Results
- We understand that greater detail regarding the net state of immunosuppression for each individual would be interesting and important in determining the pre-test probability of PCP. Given the study included all cases with a positive PCP PCR on lower respiratory specimens, we didn’t delve into the intricate detail for each patient. Given data was collected over a long period, primarily from discharge summaries and medication prescriptions unfortunately we could not accurately ascertain this level of detail consistently for all participants. In terms of chronic lung disease, we have elaborated on what chronic lung disease included (pulmonary fibrosis or COPD) in the methods. Again given the host factor is not taken into account in terms of our definitions, we have not linked information such as the present of having diabetes and on steroids in the same patient.
- We have modified the wording for describing Figure 4A and B. To avoid any confusion we have removed the P value. The confidence interval for AUC is still presented in the body of the text.
Discussion
- Line 257: adjuvant changed to adjunctive.
- We have discussed the limitation regarding the lack of cytology data and benefit it could have added to the study.
- We have added further discussion regarding the definitions used to clearly state that are different from the EORTC definitions.
Round 2
Reviewer 2 Report
Comments and Suggestions for Authors
Line 48 - the authors were discussing EORTC/MSGERC criteria and appropriately used EORTC/MSERGC terms prior to the revision. However in the revision they have inappropriately revised these words to the category descriptors used to their study even though line 48 refers to the EORTC/MSERGC criteria, not the categories the authors have defined.
Regarding the revision to category names in the rest of the paper (proven/probable now revised to definite/possible), the authors have not avoided the confusion I pointed out in the original review because their choice of a revised category label "possible" still repurposes a category label in the EORTC/MSERGC criteria that does not share the definition of that category used in this study. The authors should come up with a completely unique term that does not overlap with EORTC/MSERGC criteria, because re-defining an established term will cause confusion for the reader familiar with the consensus definitions.
Line 49 "jirovecii" is misspelled.
The authors have declined to make some recommended revisions and it appears this is primarily because limited details on the immune status of the patients are available. I understand that the inclusion criteria for their study was positive Pneumocystis PCR, but it this does not avoid the fact that most clinicians would interpret that result differently in a truly immunosuppressed individual compared to an individual without an immunocompromising condition. In fact, it appears not all of the patients are immunosuppressed but it is difficult to tell how many in the cohort are not immunosuppressed. This limitation is partially addressed in discussion by mentioning issues of generalizability and subgroup heterogeneity (line 265). But I think the lack of detailed data on immunocompromised state should be stated more specifically as a limitation. The study is too small to do a subgroup analysis of only individuals with overt immunosuppression, but one of the lingering questions with their analysis is whether more clear delineation of true infection from colonization would be achieved by Ct value if the analysis was limited to a truly immunosuppressed cohort.
I am also confused why the authors can separately report the proportion of patients with diabetes and the proportion receiving significant doses of steroids but cannot report the proportion with both as stated in their response to reviewers. I recommend adding rows for "Immunosuppressive medications" and "Steroids ≥ 2 wks > 15mg/day" to Figure 1.
